# Prevalence of hypertension and possible risk factors of hypertension unawareness among individuals aged 30–75 years from two Panamanian provinces: Results from population-based cross-sectional studies, 2010 and 2019

**Angela Isabel Del Rio** [1,2]*, **Ilais Moreno Velásquez**[1,3], **Reina Roa**[1,2], **Roger Montenegro Mendoza**[1], **Jorge Motta**[1], **Hedley K. Quintana**[1]

**1** Gorgas Memorial Institute for Health Studies, Panama City, Panama, **2** Ministry of Health, Panama City, Panama, **3** Max-Delbrück-Center for Molecular Medicine in the Helmholtz Association (MDC), Molecular Epidemiology Research Group, Berlin, Germany

* a.isabel.dr94@gmail.com

## Abstract

### Background

Recent estimates of hypertension in Panama remain unknown. We aim to describe the variation in prevalence and unawareness of hypertension in two Panamanian provinces using two different cross-sectional population-based studies and to investigate risk factors associated with hypertension unawareness.

### Methods

Data were derived from a sub-national study conducted in the provinces of Panama and Colon (PREFREC-2010 [2,733 participants]) and from a nationally representative study (ENSPA-2019), in which we restricted our analyses to the same provinces (4,653 participants). Individuals aged 30–75 years who had (a) self-reported history of hypertension or (b) blood pressure (BP) ≥140/90mmHg or (c) a combination or both were classified as hypertensive. Participants with BP≥140/90mmHg who denied a history of hypertension were considered unaware of the condition. Multivariable logistic regression models were used to estimate the association between risk factors and unawareness, expressed as odds ratios (OR) and 95% confidence interval (CI).

### Findings

In 2010, the prevalence and unawareness of hypertension in men were 51.6% (95% CI: 45.7–57.5) and 32.3% (25.4–40.1), respectively, and in women 46.0% (42.1–49.9) and 16.1% (12.6–20.4), respectively. In 2019, the prevalence and unawareness of hypertension in men were 46.5% (42.1–51.0) and 52.3% (45.9–58.6), and in women 42.1% (39.6–44.7)

**Data Availability Statement:** The datasets used and analyzed in this study contain potentially identifying information and sensitive information that is protected via Law 89/2019 as well as a mandate from the Ethics Committee. They are available on request via email to Planning directorate of the Ministry of Health at cnino@gorgas.gob.pa.

**Funding:** This study was supported by the Ministry of Health of Panama, which provided resources from the selective excise tax on cigarettes and other tobacco products. The funder had no role in study design, data collection and analysis, decision to publish, or preparation of the manuscript. There was no additional external funding received for this study.

**Competing interests:** The authors have declared that no competing interests exist.

and 33.3% (29.8–37.0). Men (2010 and 2019), age <50 years (2010 and 2019), having no/primary education (2010), and living in a non-urban region (2019) were positively associated with hypertension unawareness, whereas obesity (2010), physical inactivity (2010), family history of hypertension (2019), and BP assessment in the year before study enrollment (2010 and 2019) were inversely associated with hypertension unawareness.

## Interpretation

Benefits of a decrease in the prevalence of hypertension are being undermined by an increase in hypertension unawareness. Actions should be encouraged to strengthen the implementation of the existing healthcare program for cardiovascular risk factor control.

## Introduction

Cardiovascular disease (CVD) is a major global health concern, causing more deaths than all other causes combined [1]. CVD was responsible for 77.0% of deaths in the Americas in 2000, a percentage that increased to 81.0% in 2016 [2]. Due to this burden of disease, strategies targeting social, economic, and political determinants, as well as control of major cardiovascular risk factors, need to be implemented.

Hypertension is one of the most important modifiable risk factors for premature CVD, with most of the increase occurring in low- and middle-income countries (LMICs) [3, 4]. Variation in hypertension risk factors, such as obesity, high-sodium diet, excessive alcohol consumption, and physical inactivity, may explain some of the CVD heterogeneity between countries [3]. In addition, most people with hypertension do not have symptoms, making screening for its detection necessary [5]. Studies from 2000–2010 have shown a substantial increase in hypertension awareness in high-income countries (HIC), whereas in LMICs the increase has been slight [6]. However, since 2010, a plateau and even a decrease in hypertension awareness has been identified in HIC [7–9], as well as in LMICs [10, 11].

Panama, an upper-middle-income country [12] and the second fastest-growing country in the Americas region in terms of gross domestic product (GDP) per capita [13], faces great disparities in the distribution of wealth (Gini Index of 49.8) [14]. These disparities are also observed in health conditions among ethnic groups, as well as in different geographic areas [15–17]. The country has experienced a demographic and epidemiological transition associated with a double burden of disease (non-communicable diseases (NCDs) and infectious diseases) [18]. However, NCD-related risk factors such as hypertension predominates with a prevalence among adults of 42.3% (according to the National Health Survey of Panama (ENSPA) in 2019) [19]. Furthermore, stroke and ischemic heart disease (IHD) were the leading causes of mortality in 2019 (41.9 per 100,000 population and 41.8 per 100,000 population, respectively) [20].

Starting in 2014, a healthcare program was implemented, which focused on a multipronged approach to NCDs control, including promotion, prevention, and early detection of hypertension and other cardiovascular risk factors [21]. However, there are no recent data on the country's prevalence of hypertension to estimate the effectiveness of this healthcare program.

We aim (1) to describe the variation in the prevalence and unawareness of hypertension in two Panamanian provinces in two different population-based cross-sectional studies, conducted in 2010 and in 2019, and (2) to investigate the possible risk factors associated with hypertension unawareness.

## Materials and methods

### Study population

**The PREFREC (2010) study.** The PREFREC study (Spanish language for "Prevalence of Cardiovascular Risk Factors associated with CVD"), n = 3,590, is a sub-national, cross-sectional, descriptive study conducted between October 2010 and January 2011, designed to estimate the prevalence of well-known risk factors associated with CVDs in the provinces of Panama and Colon, where 57.4% of the total country population resided when the survey was implemented. The study included citizens older than 18 years who lived permanently in private housing of urban, rural, and indigenous areas. A complex sampling technique (three-stages, stratified, and randomized) was used. Further details regarding PREFREC have been described earlier [22]. In the present study, we included participants aged between 30 and 75 years (n = 2,733), out of them 67.5% were women (Fig 1).

**The ENSPA (2019) study.** The ENSPA study (Spanish language for "National Health Survey of Panama"), n = 28,483, is a cross-sectional, nationwide population-based study conducted between June and December 2019, designed to investigate the population's general health status and disease conditions. The study included individuals of all ages who resided permanently in private housing in rural, urban, and indigenous areas (0–14 years: n = 10,486; ≥15 years: n = 17 997). The sample design was a three-stage, stratified, and by conglomerates. The representativeness of the results is at the district level (second-level administrative division) in the entire country, except in the Panama and San Miguelito districts (province of Panama) where it has representativeness of corregimiento (third-level administrative division). Further details regarding ENSPA are described on its website in the Spanish language [23]. For the present study, we included participants living in the provinces of Panama and Colon whose age was between 30 and 75 years, (n = 4,653) (Fig 1).

Both studies were conducted by the Gorgas Memorial Institute of Health Research, the Ministry of Health of Panama (MoH), and the INEC (Spanish language for "National Institute of Statistics and Census"). The sampling design was calculated using population projections

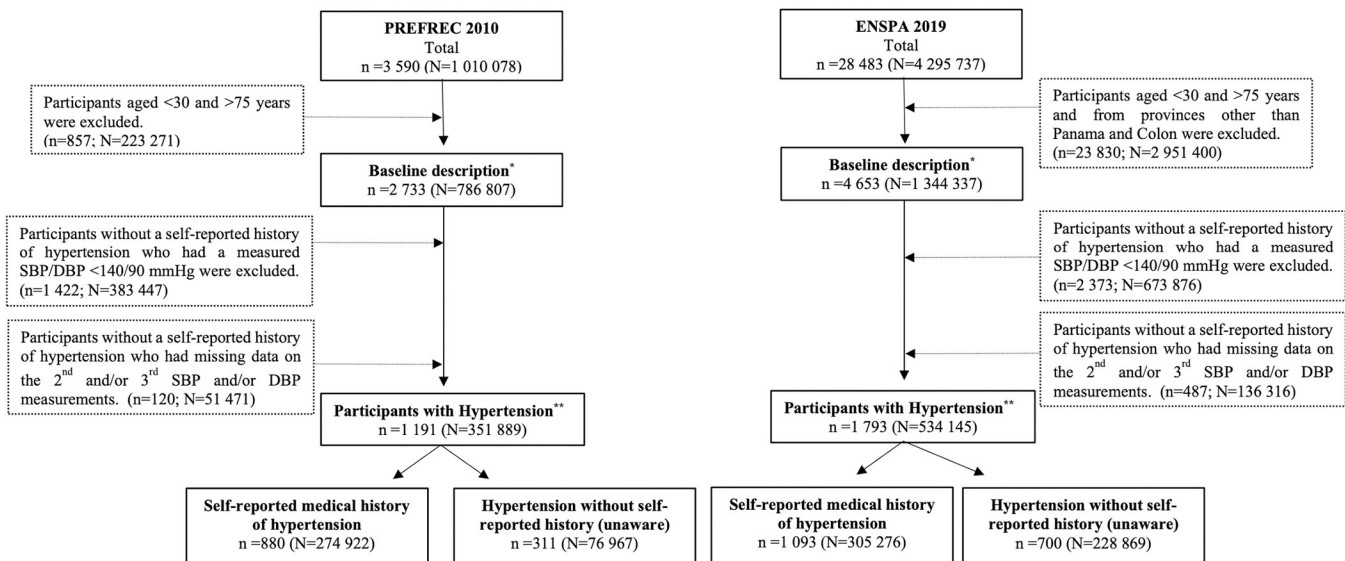

**Fig 1. Flow chart of selected participants.** SBP = systolic blood pressure. DBP = diastolic blood pressure. *Denominator for the prevalence of hypertension. **Numerator for the prevalence of hypertension and the denominator for the prevalence of hypertension unawareness.

from the two latest National Censuses (2000 for the PREFREC (2010) study and 2010 for the ENSPA (2019) study) for both studies when the data collection took place [24].

## Data collection

Participants answered a questionnaire in Spanish through in-person interviews, which collected information on demographics, socioeconomic, medical and family history, lifestyle, and anthropometric measurements (including blood pressure). Weight, height, and blood pressure (BP) were undertaken using standardized instruments.

## Outcome assessment

Outcome variables were assessed using self-reported medical history of hypertension and BP measurements.

Participants' self-reported medical history was based on the question (presented in both surveys): *have you ever been told by a physician that you have hypertension, also called high blood pressure?* (The fragment, *also called high blood pressure* was solely present in the PREFREC study).

Participants' BP measurements were considered to be the average of the second and third BP measurement. In the PREFREC (2010) study, BP measurements have been previously described [25]. In brief, systolic blood pressure (SBP) and diastolic blood pressure (DBP) were measured three times in a sitting position after a minimum of five minutes of rest and recorded with a five-minute interval between each other, using an electronic device (American Diagnostic Corporation model 6013). Measurements were performed in the right arm. In the ENSPA (2019) study, following the WHO Stepwise Approach to Surveillance (STEPS) [26], SBP and DBP were measured three times in a sitting position after fifteen minutes of rest and recorded with a three-minute interval between each other, using an electronic device (OMRON model HEM-7120). Measurements were performed in the left arm.

Hypertension was defined as (a) having a self-reported medical history of hypertension or (b) having a mean SBP ≥140 and/or DBP ≥90 mmHg (regardless of their self-reported medical history of hypertension) or (c) a combination of both [27]. This sample constituted our study population for estimating the prevalence of hypertension unawareness. Unawareness of hypertension was defined as having a mean SBP ≥140 and/or DBP ≥90 mmHg and no self-reported medical history of hypertension. We excluded participants who denied self-reported medical history of hypertension and were also missing the second and/or third BP measurement(s) (n = 120 in the PREFREC study; n = 487 in the ENSPA study) (Fig 1).

## Exposure variables

Demographic and socioeconomic characteristics included age (years), sex (men/women), ethnicity (Afro-Panamanian, Caucasian, Mestizo, Indigenous, or others), region (urban, non-urban), education (no/primary education, secondary education, and higher education), and monthly family income (MFI) (<250 Panamanian balboas (PAB), 250–999 PAB, and ≥1,000 PAB).

A wide range of established risk factors for hypertension was evaluated. Smoking tobacco consumption was classified into current smokers, ex-smoker, or non-smokers. Body mass index (BMI) was obtained by dividing the participant's weight measurement in kilograms by the square of their height in meters squared and categorized into underweight (<18.5 kg/m$^2$), normal weight (18.5 to 24.9 kg/m$^2$), overweight (25 to 29.9 kg/m$^2$), and obesity (≥30 kg/m$^2$) [28]. Physical inactivity was assessed in the PREFREC (2010) study when they reported doing less than 150 minutes of certain activities per week. In the ENSPA (2019) study, physical

inactivity among study participants was defined if the metabolic equivalents per minute (METs-minute) per week calculated using the GPAQ (General Practice Assessment of Quality) were less than 600 [29]. Family history of hypertension was recorded when first- and/or second-degree relatives were reported to have the condition. Self-reported medical history of diabetes was defined if the participant answered positively to the question, *have you been told by a physician that you have diabetes?*

Healthcare-related factors like BP assessment in the year before study enrollment (assessed in both studies) and health check-up in the year before study enrollment (assessed in the ENSPA (2019) study) were evaluated. The S1 Table shows the definitions of the exposure variables used in the present study.

## Statistical analysis

Continuous variables are presented as median and interquartile range (IQR). Categorical variables are presented as percentages with the corresponding 95% confidence intervals (CIs). Both categorical and continuous variables assessed the total population of the provinces of Panama and Colon when the respective study took place using the expansion weights calculated and provided by the INEC. Results are then presented weighted (N). Missing data were excluded from the analysis.

To examine the association of each exposure variable with hypertension unawareness, its respective odds ratios (OR) with their 95% CIs were calculated by applying unconditional logistic regression models. Crude and multivariable logistic regression models were performed to address potential confounding bias. The first adjusted model (model A) accounted for the demographic factors (sex, age, and ethnicity), established risk factors for hypertension (tobacco consumption, BMI categories, physical inactivity, family history of hypertension, and self-reported medical history of diabetes), and BP assessment in the year before study enrollment. Then, in a second model (model B) the socioeconomic determinants (region, MFI, and education) were added to model A. In addition, we included in the (S2 and S3 Tables) crude and multivariable logistic regression models using merged data from both studies (both PREFREC and ENSPA) in which the prevalence of hypertension (S2 Table) and the prevalence of hypertension unawareness (S3 Table) were the outcome variables, respectively, with the aforementioned predictor variables together with the study period as an additional predictor (being PREFREC the reference level).

Prevalence tables are presented stratified by sex. Data were analyzed using SPSS V.20.0 software and R program version 4.0.0 with the survey package version 4.0.

## Ethical statement

Studies were approved by the National Bioethics Committee of the Gorgas Memorial Institute of Health Studies and conducted following the Declaration of Helsinki. All participants signed an informed consent to be enrolled in the study.

## Results

### Demographic and socioeconomic characteristics

Overall, the predominant ethnicity was mestizo (men: 60.4% in PREFREC (2010) and 47.5% in ENSPA (2019), women: 60.7% in PREFREC (2010) and 49.4% in ENSPA (2019)), and most individuals lived in an urban region (Table 1). According to the education levels, fewer participants reported having no education/primary education in ENSPA (2019), compared to PREFREC (2010) (26.8% vs 20.3% for men; 32.1% vs 23.0% for women). In contrast, more

**Table 1. Distribution of baseline characteristics in individuals aged between 30 and 75 years by sex and the study year (PREFREC 2010 and ENSPA 2019) in the provinces of Panama and Colon.**

| Demographic and socioeconomic characteristics | Men | | Women | |
|---|---|---|---|---|
| | PREFREC 2010 | ENSPA 2019 | PREFREC 2010 | ENSPA 2019 |
| | (N = 255 481) | (N = 666 752) | (N = 531 325) | (N = 677 584) |
| Age, years | | | | |
| Median (IQR) | 52.0 (40.0–62.0) | 50.0 (40.0–61.0) | 48.0 (39.0–58.0) | 48.0 (38.0–60.0) |
| Ethnicity–% (95% CI) | | | | |
| Mestizo | 60.4 (54.7–65.9) | 47.5 (42.4–51.6) | 60.7 (57.0–64.2) | 49.4 (46.9–51.8) |
| Afro-Panamanian | 17.9 (13.9–22.6) | 24.8 (21.2–28.9) | 20.4 (17.7–23.3) | 20.5 (18.6–22.5) |
| Caucasian | 12.7 (9.3–17.0) | 21.2 (17.9–24.9) | 14.0 (11.5–16.8) | 21.1 (19.2–23.3) |
| Indigenous | 2.8 (1.3–6.2) | 4.3 (3.0–6.0) | 3.1 (2.0–4.8) | 5.3 (4.2–6.7) |
| Others | 6.3 (3.9–10.0) | 2.2 (1.3–3.7) | 1.9 (1.1–3.5) | 3.6 (2.8–4.8) |
| Region–% (95% CI) | | | | |
| Urban | 85.3 (82.9–87.4) | 79.5 (76.5–82.2) | 87.6 (86.2–88.9) | 82.0 (80.3–83.5) |
| Non-urban | 14.7 (12.6–17.1) | 20.5 (17.8–23.5) | 12.4 (11.1–13.8) | 18.0 (16.5–19.7) |
| Education–% (95% CI) | | | | |
| No/primary education | 26.8 (22.4–31.8) | 20.3 (17.4–23.4) | 32.1 (28.7–35.7) | 23.0 (21.0–25.0) |
| Secondary education | 44.5 (38.8–50.2) | 58.4 (54.2–62.4) | 46.7 (42.9–50.5) | 55.0 (52.5–57.4) |
| Higher education | 28.7 (23.6–34.5) | 21.4 (18.0–25.1) | 21.3 (18.4–24.5) | 22.1 (20.1–24.2) |
| Monthly Family Income–% (95% CI) | | | | |
| <250 PAB | 19.3 (15.9–23.8) | 19.2 (16.3–22.5) | 31.7 (28.4–35.3) | 23.9 (22.0–26.0) |
| 250–999 PAB | 64.3 (58.8–69.3) | 58.6 (54.4–62.6) | 55.5 (51.7–59.3) | 61.0 (58.6–63.4) |
| ≥1,000 PAB | 16.2 (12.4–20.8) | 22.2 (18.8–26.0) | 12.7 (10.3–15.7) | 15.1 (13.3–17.0) |
| **Established risk factors for hypertension–% (95% CI)** | | | | |
| Tobacco consumption | | | | |
| Current smoker | 11.7 (8.6–15.8) | 7.6 (5.7–10.0) | 3.9 (2.6–5.7) | 2.4 (1.7–3.3) |
| Ex-smoker | 47.0 (41.3–52.7) | 4.2 (2.9–6.2) | 15.7 (13.1–18.6) | 1.0 (0.6–1.6) |
| Non-smoker | 41.3 (35.8–47.0) | 88.2 (85.3–90.6) | 80.5 (77.3–83.3) | 96.6 (95.6–97.4) |
| BMI categories[1] | | | | |
| Underweight | 1.4 (0.5–3.7) | 1.4 (0.7–2.6) | 1.1 (0.6–1.9) | 1.5 (1.0 – 2.3) |
| Normal weight | 40.7 (35.2–46.5) | 23.6 (20.2–27.4) | 26.3 (23.1–29.7) | 18.3 (16.4–20.4) |
| Overweight | 32.7 (27.7–38.1) | 41.1 (36.8–45.6) | 37.3 (33.7–41.1) | 31.1 (28.8–33.5) |
| Obesity | 25.2 (20.5–30.5) | 33.8 (29.7–38.2) | 35.3 (31.8–39.0) | 49.1 (46.5–51.7) |
| Physical inactivity | 20.4 (16.4–25.1) | 52.5 (48.2–56.7) | 12.3 (10.1–15.0) | 66.9 (64.4–69.3) |
| Family history of hypertension | 64.0 (58.2–69.4) | 37.1 (33.2–41.1) | 74.4 (71.0–77.5) | 46.0 (43.5–48.4) |
| Self-reported medical history of diabetes | 10.1 (7.3–13.8) | 4.9 (3.5–6.8) | 9.5 (7.5–11.9) | 8.6 (7.3–10.2) |
| **Hypertension–% (95% CI)** | | | | |
| Yes | 51.6 (45.7–57.5) | 46.5 (42.1–51.0) | 46.0 (42.1–49.9) | 42.1 (39.6–44.7) |
| Hypertension unawareness[2] | 32.3 (25.4–40.1) | 52.3 (45.9–58.6) | 16.1 (12.6–20.4) | 33.3 (29.8–37.0) |
| **Healthcare-related factors–% (95% CI)** | | | | |
| BP assessment in the year before study enrollment | 73.9 (68.7–78.5) | 34.9 (31.0–39.0) | 79.8 (76.6–82.7) | 40.2 (37.8–42.6) |
| At least a health check-up in the year before study enrollment | NA | 57.2 (53.0–61.2) | NA | 68.5 (66.2–70.8) |
| BP assessment[3] | NA | 48.2 (42.7–53.7) | NA | 49.6 (46.7–52.6) |

N = weighted study population. IQR = interquartile range. PAB = Panamanian balboa. BP = blood pressure. NA = not applicable. BMI = body mass index.

MFI = monthly family income. SBP = systolic blood pressure. DBP = diastolic blood pressure. % = percentages. CIs = confidence intervals.

[1] According to World Health Organization (WHO).

[2] SBP/DBP ≥140/90 mmHg and no self-reported medical history of hypertension.

[3] BP assessment among those with at least a health check-up in the year before study enrollment. Missing data on the PREFREC 2010 study: BP measurements (N = 83 177; 11%), MFI (N = 8 675; 7.8%). Missing data on the ENSPA 2019 study: BP measurements (N = 167 088; 12%), MFI (N = 65 961; 5%), physical inactivity (N = 259 434; 19%), BMI (N = 164 047; 12%).

participants reported having secondary education (an increase of 13.9 percentage points for men and 8.3 percentage points for women). Fewer men reported having higher education (28.7% vs 21.4%), while a slight increase was present in women (21.3% vs 22.1%) (Table 1). Regarding MFI, fewer women reported MFI of <250 PAB in the ENSPA (2019), compared to the PREFREC (2010) (31.7% vs 23.9%), whereas MFI of 250–999 PAB and MFI of ≥1,000 PAB increased by 8.3% and 2.4%, respectively. Among men, MFI of <250 PAB remained unchanged (20.0%), MFI of 250–999 PAB decreased by 5.7%, and MFI of ≥1,000 PAB increased by 6.0% (Table 1).

## Established risk factors for hypertension

The prevalence of current and ex-smokers in PREFREC (2010) was 11.7% and 47.0% in men, respectively, and among women 3.9% and 15.7%, respectively. In the ENSPA (2019) study, the prevalence of current smokers was 7.6% for men and 2.4% for women, whereas the prevalence of ex-smokers was 4.2% in men and 1.0% in women. Regarding BMI, we found that the prevalence of overweight in men was 32.7% (95% CI: 27.7–38.1) in PREFREC (2010) and 41.1% (95% CI: 36.8–45.6) in ENSPA (2019). In contrast, among women, obesity was present in 35.3% (95% CI: 31.8–39.0) of participants in PREFREC (2010) and in 49.1% (95% CI: 46.5–51.7) in ENSPA (2019). Physical inactivity in PREFREC (2010) was present in 20.4% and 12.3% of men and women, respectively, and in ENSPA (2019), it accounted for 52.5% of men and 66.9% of women (Table 1).

## Hypertension

In the PREFREC (2010), the prevalence of hypertension and hypertension unawareness in men were 51.6% (95% CI: 45.7–57.5) and 32.3% (95% CI: 25.4–40.1), respectively, and in women 46.0% (95% CI: 42.1–49.9) and 16.1% (95% CI: 12.6–20.4), respectively (Table 1 and Fig 2). In the ENSPA (2019), the prevalence of hypertension and hypertension unawareness in men were 46.5% (95% CI: 42.1–51.0) and 52.3% (95% CI: 45.9–58.6), respectively, and in women 42.1% (95% CI: 39.6–44.7) and 33.3% (95% CI: 29.8–37.0), respectively (Table 1 and Fig 2). As shown in the S2 and S3 Tables, after merging all data from both studies and adjusting for all other variables, there was a 21% borderline decrease in the odds of having hypertension in the ENSPA (2019) study compared to the PREFREC study (ENSPA OR: 0.79; 95% CI: 0.62–1.02), but a 70% increase in the odds of being unaware (ENSPA OR: 1.70; 95% CI: 1.15–2.51), respectively.

## Healthcare-related factors

We found that fewer men reported having a BP assessment in the year before study enrollment in the ENSPA (2019) study compared to the PREFREC (2010) study (73.9% [95% CI: 68.7–78.5] in PREFREC (2010) vs 34.9% [95% CI: 31.0–39.0] in ENSPA (2019)). Similarly, women displayed a decrease from 79.8% (95% CI: 76.6–82.7) in PREFREC (2010) to 40.2% (95% CI: 37.8–42.6) in ENSPA (2019) (Fig 3). Additionally, in ENSPA (2019), 57.2% of men and 68.5% of women had a health check-up in the year before study enrollment. Amongst them, 48.2% of men and 49.6% of women reported having their BP assessed the year before study enrollment.

## Risk factors associated with unawareness of hypertension

Table 2 shows the crude and adjusted analysis for the associations between each exposure variable and hypertension unawareness, stratified by the study year. Regarding demographic factors, men were more likely to be unaware, compared with women, in the PREFREC (2010)

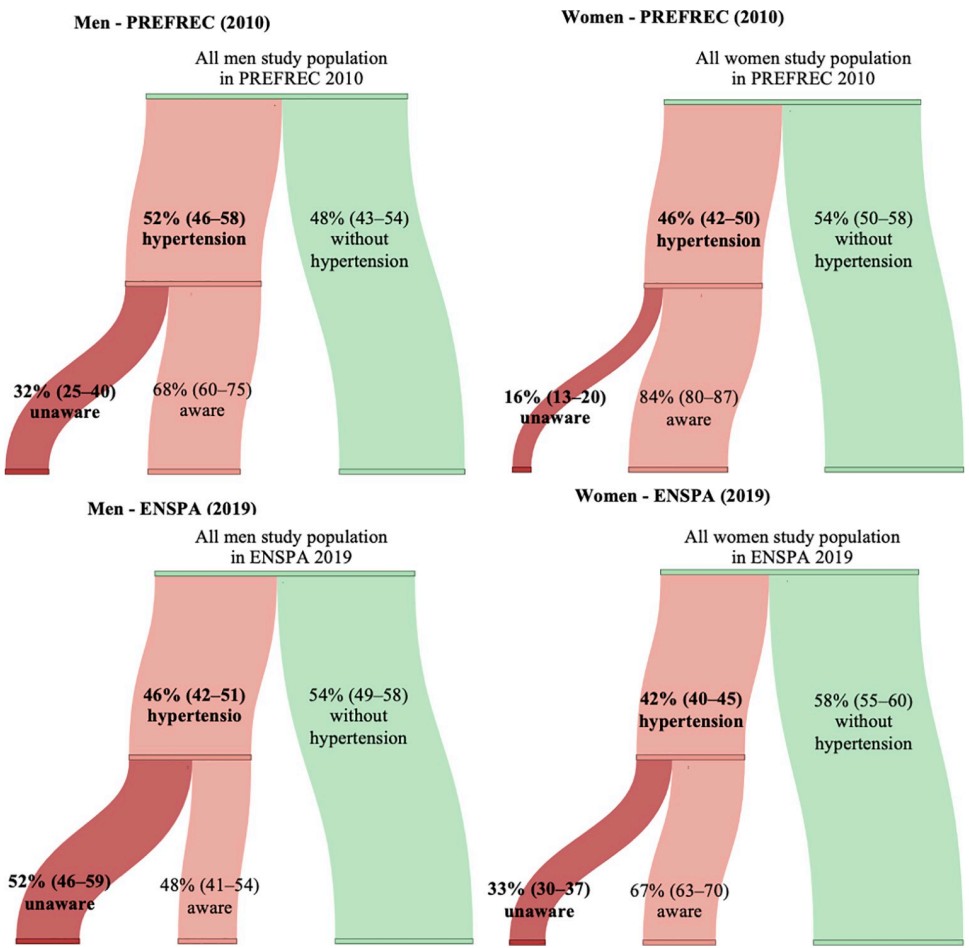

**Fig 2. Prevalence and unawareness of hypertension stratified by sex and the study year.** BP = blood pressure. The PREFREC (2010) study: men (n = 834 [N = 255 481]), women (n = 1 899 [N = 531 326]). The ENSPA (2019) study: men (n = 1 317 [N = 666 751]), women (n = 3 334 [N = 677 584]).

(OR: 2.31; 95% CI: 1.29–4.12) and in the ENSPA (2019) (OR: 2.09; 95% CI: 1.37–3.17) (model B in Table 2). Whereas individuals aged <50 years were more likely to be unaware, compared with those aged ≥50 years, in PREFREC (2010) (OR: 1.84; 95% CI: 1.06–3.17) and in ENSPA (2019) (OR: 2.43; 95% CI: 1.57–3.76) (model B in Table 2). No association with ethnicity was found (S4 and S5 Tables).

When considering hypertension risk factors, obesity was associated with lower odds of having hypertension unawareness, compared to a normal weight, in PREFREC (2010) (OR: 0.48; 95% CI: 0.25–0.91), but not in ENSPA (2019) (OR: 0.76; 95% CI: 0.44–1.29) (model B in Table 2). Similarly, physical inactivity was associated with decreased odds of having hypertension unawareness in PREFREC (2010) (OR: 0.43; 95% CI: 0.22–0.84), but not in ENSPA (2019) (OR: 0.88; 95% CI: 0.57–1.36) (model B in Table 2). Family history of hypertension had a borderline inverse association for hypertension unawareness in PREFREC (2010) (OR: 0.62; 95% CI: 0.36–1.06) and an inverse association in ENSPA (2019) (OR: 0.18; 95% CI: 0.11–0.27) (model B in Table 2). No association was found with tobacco consumption regardless of the study year, and self-reported medical history of diabetes had a borderline inverse association with hypertension unawareness in ENSPA (2019) (OR: 0.55; 95% CI: 0.31–1.00) (model B in Table 2).

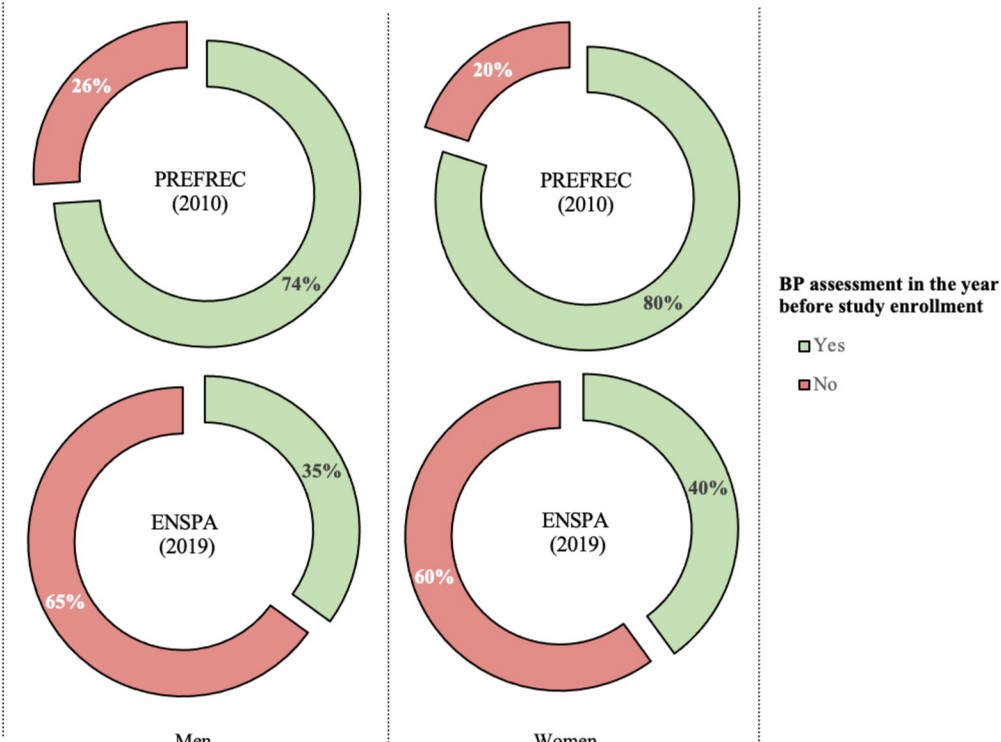

**Fig 3. BP assessment in the year before study enrollment stratified by sex and the study year.** BP = blood pressure. The PREFREC (2010) study: men (n = 834 [N = 255 481]), women (n = 1 899 [N = 531 326]). The ENSPA (2019) study: men (n = 1 317 [N = 666 751]), women (n = 3 334 [N = 677 584]).

Furthermore, having a BP assessment in the year before study enrollment was associated with decreased odds of being unaware in PREFREC (2010) (OR: 0.46; 95% CI: 0.25–0.87) and in ENSPA (2019) (OR: 0.35; 95% CI: 0.23–0.53) (model B in Table 2).

When examining the association between socioeconomic factors and unawareness, in PRE-FREC (2010), higher odds of having hypertension unawareness was found with having no/primary education (OR: 2.27; 95% CI: 1.06–4.86), compared to higher education, and in ENSPA (2019), with living in a non-urban region (OR: 1.63; 95% CI: 1.03–2.60), compared to living in an urban region (model B in Table 2).

## Discussion

Our findings indicate a borderline decrease in the odds of having hypertension between the PREFREC (2010) and ENSPA (2019) studies, but an increase in the odds of being unaware of hypertension. Further, there was an increase in the prevalence of overweight and obesity in the provinces of Panama and Colon.

Panama has experienced exceptional economic growth over the past three decades, evidenced by the increase in GDP per capita, which has doubled in the last ten years [12]. This growth, in turn, has produced a rapidly expanding urbanization associated with increased migration from rural to urban areas [30, 31], which has augmented the number of people exposed to lifestyle factors related to hypertension, such as westernized dietary habits, excess sodium in the diet, harmful alcohol consumption, stress, and physical inactivity [32–34]. It is likely that these factors have contributed to the increase in the prevalence of obesity in Panama over the past three decades (from 3.8% in men and 7.6% in women, in 1982, to 16.9% in men

**Table 2. Crude and adjusted logistic regression analysis of hypertension unawareness (SBP/DBP ≥140/90 mmHg without a self-reported medical history of hypertension) in hypertensive individuals aged between 30 and 75 years by the study year (PREFREC 2010 and ENSPA 2019) in the provinces of Panama and Colon.** Odds ratio (OR) and 95% confidence intervals (CIs).

| | PREFREC 2010 | | | ENSPA 2019 | | |
|---|---|---|---|---|---|---|
| | Crude analysis OR (95% CI) | Adjusted analysis OR (95% CI) | | Crude analysis OR (95% CI) | Adjusted analysis OR (95% CI) | |
| | | Model A | Model B | | Model A | Model B |
| **Demographic Factors** | | | | | | |
| Sex | | | | | | |
| Women (reference) | 1 (ref) | 1 (ref) | 1 (ref) | 1 (ref) | 1 (ref) | 1 (ref) |
| Men | **2.49** | **2.30** | **2.31** | **2.19** | **2.13** | **2.09** |
| | **(1.60–3.89)** | **(1.33–3.96)** | **(1.29–4.12)** | **(1.62–2.97)** | **(1.42–3.20)** | **(1.37–3.17)** |
| Age group | | | | | | |
| ≥50 years (reference) | 1 (ref) | 1 (ref) | 1 (ref) | 1 (ref) | 1 (ref) | 1 (ref) |
| <50 years | **1.69** | **1.68** | **1.84** | **2.22** | **2.29** | **2.43** |
| | **(1.07–2.65)** | **(1.01–2.80)** | **(1.06–3.17)** | **(1.60–3.09)** | **(1.54–3.40)** | **(1.57–3.76)** |
| **Established risk factors for hypertension** | | | | | | |
| Tabaco consumption | | | | | | |
| Non-smoker | 1 (ref) | 1 (ref) | 1 (ref) | 1 (ref) | 1 (ref) | 1 (ref) |
| Ex-smoker | 1.28 | 0.90 | 0.81 | 0.89 | 1.34 | 1.24 |
| | (0.78–2.10) | (0.50–1.62) | (0.45–1.47) | (0.37–2.12) | (0.38–4.72) | (0.33–4.68) |
| Current smoker | 1.64 | 0.56 | 0.56 | 1.53 | 0.94 | 0.73 |
| | (0.66–4.10) | (0.16–1.95) | (0.16–1.97) | (0.68–3.42) | (0.34–2.58) | (0.27–1.98) |
| Body mass index categories[1] | | | | | | |
| Underweight | 3.38 | 2.62 | 2.36 | 0.69 | 0.89 | 0.88 |
| | (0.88–12.98) | (0.52–13.11) | (0.46–12.01) | (0.11–4.49) | (0.24–3.35) | (0.24–3.32) |
| Normal weight | 1 (ref) | 1 (ref) | 1 (ref) | 1 (ref) | 1 (ref) | 1 (ref) |
| Overweight | 0.70 | 0.64 | 0.63 | 0.96 | 1.09 | 1.16 |
| | (0.41–1.18) | (0.37–1.09) | (0.36–1.10) | (0.60–1.54) | (0.65–1.85) | (0.68–1.98) |
| Obesity | **0.45** | **0.45** | **0.48** | **0.60** | 0.79 | 0.76 |
| | **(0.25–0.80)** | **(0.24–0.84)** | **(0.25–0.91)** | **(0.39–0.94)** | (0.47–1.34) | (0.44–1.29) |
| Physical inactivity | | | | | | |
| No (reference) | 1 (ref) | 1 (ref) | 1 (ref) | 1 (ref) | 1 (ref) | 1 (ref) |
| Yes | **0.54** | **0.47** | **0.43** | 0.95 | 0.92 | 0.88 |
| | **(0.29–0.99)** | **(0.25–0.88)** | **(0.22–0.84)** | (0.68–1.33) | (0.61–1.39) | (0.57–1.36) |
| Family history of hypertension | | | | | | |
| No (reference) | 1 (ref) | 1 (ref) | 1 (ref) | 1 (ref) | 1 (ref) | 1 (ref) |
| Yes | **0.51** | 0.60 | 0.62 | **0.17** | **0.18** | **0.18** |
| | **(0.31–0.83)** | (0.35–1.00) | (0.36–1.06) | **(0.11–0.24)** | **(0.12–0.27)** | **(0.11–0.27)** |
| Self-reported medical history of diabetes | | | | | | |
| No (reference) | 1 (ref) | 1 (ref) | 1 (ref) | 1 (ref) | 1 (ref) | 1 (ref) |
| Yes | 0.85 | 1.00 | 1.09 | **0.31** | 0.57 | 0.55 |
| | (0.45–1.60) | (0.51–2.00) | (0.54–2.19) | **(0.18–0.54)** | (0.32–1.02) | (0.31–1.00) |
| **BP assessment in the year before study enrollment** | | | | | | |
| No (reference) | 1 (ref) | 1 (ref) | 1 (ref) | 1 (ref) | 1 (ref) | 1 (ref) |
| Yes | **0.31** | **0.48** | **0.46** | **0.25** | **0.35** | **0.35** |
| | **(0.18–0.54)** | **(0.27–0.86)** | **(0.25–0.87)** | **(0.18–0.36)** | **(0.23–0.52)** | **(0.23–0.53)** |
| **Socioeconomic Factors** | | | | | | |
| Region | | | | | | |
| Urban (reference) | 1 (ref) | | 1 (ref) | 1 (ref) | | 1 (ref) |

*(Continued)*

**Table 2.** (Continued)

| | PREFREC 2010 | | | ENSPA 2019 | | |
|---|---|---|---|---|---|---|
| | Crude analysis OR (95% CI) | Adjusted analysis OR (95% CI) | | Crude analysis OR (95% CI) | Adjusted analysis OR (95% CI) | |
| | | Model A | Model B | | Model A | Model B |
| Non-urban | 1.41 | | 0.94 | **1.78** | | **1.63** |
| | (0.99–2.01) | | (0.60–1.47) | **(1.27–2.50)** | | **(1.03–2.60)** |
| Education | | | | | | |
| Higher education | 1 (ref) | | 1 (ref) | 1 (ref) | | 1 (ref) |
| Secondary education | 0.79 | | 1.39 | 1.49 | | 1.42 |
| | (0.43–1.46) | | (0.74–2.64) | (0.94–2.37) | | (0.71–2.81) |
| No/primary education | 1.19 | | **2.27** | 1.13 | | 0.95 |
| | (0.65–2.18) | | **(1.06–4.86)** | (0.69–1.85) | | (0.45–1.99) |
| Monthly Family Income | | | | | | |
| ≥1,000 PAB | 1 (ref) | | 1 (ref) | 1 (ref) | | 1 (ref) |
| 250–999 PAB | 0.90 | | 0.69 | **1.68** | | 1.39 |
| | (0.45–1.78) | | (0.33–1.48) | **(1.08–2.65)** | | (0.68–2.81) |
| <250 PAB | 0.32 | | 0.62 | **1.69** | | 1.00 |
| | (0.42–1.70) | | (0.26–1.50) | **(1.03–2.75)** | | (0.47–2.10) |

SBP = systolic blood pressure. DBP = diastolic blood pressure. OR = odds ratio. CIs = confidence intervals. BP = blood pressure. BMI = body mass index.

PAB = Panamanian Balboa.

Model A = adjusted by sex, age group, ethnicity, BMI categories, physical inactivity, family history of hypertension, self-reported medical history of diabetes, tobacco consumption, and BP assessment in the year before study enrollment.

Model B = further adjusted by region, education, and monthly family income.

[1]According to World Health Organization (WHO).

and 23.8% in women, in 2008 [35]) and continues to rise as suggested by our study. We also found that more than half of individuals in the ENSPA (2019) study were physically inactive, a higher estimate than that reported in other countries in Latin America and the Caribbean [36].

The aforementioned findings could explain our high estimates of prevalence of hypertension compared with those reported by other countries in the Americas region [7, 9–11, 37–39]. Nevertheless, we found a decrease in the prevalence of hypertension between the PREFREC (2010) and ENSPA (2019) study, as opposed to other countries in the region, such as Peru, that reported an increasing prevalence of hypertension attributed to recent economic growth [10], whereas the prevalence in Brazil [37], Canada [7], and the United States [9] has remained unchanged. In agreement with our study, recent studies have reported a decrease in the prevalence of hypertension despite an increase in obesity [4, 11]. Our results suggest that other risk factors associated with hypertension such as salt intake, smoking, and exposure to dietary fatty acids have improved [4, 11]. For example, in the case of smoking, in 2005, Panama adopted the WHO Framework Convention on Tobacco Control. Since then, two laws were introduced: a law regulating tobacco control in 2008, and a tobacco tax increase in 2009 [40, 41]. Associated with these laws, there has been a decrease in tobacco consumption and a reduction in the incidence of acute myocardial infarction [41]. Likewise, we found an improvement in education levels: an important and well-known determinant of health [15, 32, 39, 42]. Our study suggests that as health determinants, such as education, improve and risk factors for hypertension, such as tobacco consumption, are diminished, the prevalence of hypertension as well as its potential complications are reduced.

Our results showed an increased prevalence of hypertension unawareness across both studies in the provinces of Panama and Colon. Similarly, a significant increase in the prevalence of hypertension unawareness was found in Canada [7], the United States [9], and in Peru [10]. In contrast, recent findings from other studies performed in Chile [11] and Brazil [36], showed either a plateau or a decrease in hypertension unawareness. When global trends in hypertension were examined between 1990 to 2019 in Latin America and the Caribbean, a slight decrease in hypertension unawareness was found until the mid-2000s before flattening [4]. The increase in hypertension unawareness in some countries has been attributed to reduced funding directed to hypertension programs, a fragmented healthcare system, lack of educational programs, and lack of BP screening in the younger population (18–44 years) [7, 9, 10].

Our study identified a decrease in the prevalence of BP assessment in the year before study enrollment between the PREFREC (2010) and ENSPA (2019) study. As expected, BP assessment in the year before study enrollment was a factor strongly associated with decreased odds of having hypertension unawareness in both studies, which is consistent with previous research suggesting that the frequency of BP assessment is related to awareness [4, 9, 43–45]. Furthermore, we found that approximately half of the individuals in the ENSPA (2019) study who participated in health check-ups did not have a BP assessment, pointing to a significant flaw of these health check-ups and a lack of compliance with the national healthcare program standards [21]. Of note, the current national healthcare program states that BP levels should be measured in the right arm at every health check-up in patients 18 years or older.

Several factors were associated with decreased odds of having hypertension unawareness, including female sex, older age ($\geq$50 years), obesity, physical inactivity, family history of hypertension, and BP assessment in the year before study enrollment. Women were less likely to be unaware, compared to men, a finding consistent with a large body of evidence on sex and health-seeking behaviors [39, 43, 45–51]. Older individuals ($\geq$50 years) were less likely to have hypertension unawareness compared to younger individuals (<50 years). This is likely due to a higher healthcare utilization by older individuals, as other studies have suggested [32, 39, 45–49, 51, 52]. Another factor associated with a lower risk of hypertension unawareness was having a family history of hypertension, which is consistent with previous findings [43, 46, 49, 52].

Paradoxically, in the PREFREC (2010) study, obesity and physical inactivity were found to be associated with decreased odds of having hypertension unawareness. One possible explanation may be that these groups have been targeted by screening programs and have more contact with health professional [32, 39, 44, 46–52]. On the other hand, the inverse association with hypertension unawareness was not observed in the ENSPA (2019) study, suggesting a decrease in regular health check-ups or reduced screening in this group of individuals.

Socioeconomic factors related to hypertension unawareness differed across both studies. We found in the PREFREC (2010) study that having no education/primary education and in the ENSPA (2019) study that living in a non-urban region were associated with increased odds of having hypertension unawareness. The lack of association of education and unawareness in the ENSPA (2019) study could be explained by the finding of improved educational levels, when compared to the PREFREC (2010) study. Both of these socioeconomic factors have been previously recognized in many countries [10, 32, 39, 44, 46–48, 51]. However, among Chinese men and women, having a low educational level was associated with a lower risk of being unaware [52]. Additionally, compared to other studies [47, 50], we did not find an association between household income and hypertension unawareness.

Hypertension has emerged as an important risk factor in younger populations (aged 25–49 years) [53]. A previous Panamanian study reported that hypertension was the most common risk factor associated with stroke in young adults [54]. Likewise, a prospective study from

China found that having early-onset hypertension (<45 years) increases CVD mortality compared with late-onset hypertension (≥65 years) [55]. Therefore, greater emphasis should be placed on screening the younger population, as stated in our national healthcare program [21].

Our study has several limitations. First, the PREFREC (2010) and ENSPA (2019) are cross-sectional study designs and thus, causality should not be inferred. Second, medical history of hypertension was based on self-reporting and there may have been reporting biases. Moreover, although both studies collected three BP measurements for each participant, these were taken during a single visit (diagnosis requires two or more visits) and in different arms. However, this is a common approach in large-scale epidemiological studies, and because of design and time constrains, measurements in both arms could not be assessed [10, 26, 46, 52, 56]. Third, in the two studies, physical inactivity was assessed with different methodologies, so we were not able to analyze variations across studies. Fourth, we did not perform a temporal analysis of trend over time. Finally, although our study was limited to two Panamanian provinces that contain around 60% of the country's population, the results cannot be extrapolated to the whole country. Moreover, the provinces of Panama and Colon are predominantly urban, with the province of Colon having the highest concentration of people of Afro-Panamanians, who have a higher risk of hypertension and obesity compared to other ethnic groups in the country [25, 35]. The strength of this study was the large study population as well as the sampling methodology that allowed us to provide robust information.

## Conclusions

Although the study found a reduction in the prevalence of hypertension, our results also showed a worrisome increase in hypertension unawareness. These findings may be due, in part, to a weakening of CVD prevention healthcare program, as evidenced by a decrease in BP assessment in the year before study enrollment when one compares the PREFREC (2010) and the ENSPA (2019) studies. In addition, almost half of study participants who reported having had a health check-up in the year prior to the ENSPA (2019) study, denied having had a BP assessment.

Our study points to the need of strengthening the CVD healthcare program implemented in 2014 [21]. Future research aiming to develop interventions to improve early diagnosis and reduce risk factors associated with hypertension is warranted. It is likely that if the finding related to hypertension unawareness is not promptly addressed, we could expect a rebound in the decrease in the prevalence of hypertension and the observed decline in CVD mortality [15, 57].

## Supporting information

**S1 Table. Definition of exposure variables based on the study year.**
(PDF)

**S2 Table. Crude and adjusted logistic regression analysis of hypertension prevalence (SBP/DBP ≥140/90 mmHg detected during the study and/or self-reported medical history of hypertension) in participants aged between 30 and 75 years in the provinces of Panama and Colon using merged data from the PREFREC (2010) and ENSPA (2019) studies.** Odds ratio (OR) and 95% confidence intervals (CIs).
(PDF)

**S3 Table. Crude and adjusted logistic regression analysis of hypertension unawareness (SBP/DBP ≥140/90 mmHg without a self-reported medical history of hypertension) in hypertensive participants aged between 30 and 75 years in the provinces of Panama and**

**Colon using merged data from the PREFREC (2010) and ENSPA (2019) studies.** Odds ratio (OR) and 95% confidence intervals (CIs).
(PDF)

**S4 Table. Crude and adjusted logistic regression analysis of hypertension unawareness (SBP/DBP ≥140/90 mmHg without a self-reported medical history of hypertension) in hypertensive participants aged between 30 and 75 years in the PREFREC study in the provinces of Panama and Colon.** Odds ratio (OR) and 95% confidence intervals (CIs).
(PDF)

**S5 Table. Crude and adjusted logistic regression analysis of hypertension unawareness (SBP/DBP ≥140/90 mmHg without a self-reported medical history of hypertension) in hypertensive participants aged between 30 and 75 years in the ENSPA study in the provinces of Panama and Colon.** Odds ratio (OR) and 95% confidence intervals (CIs).
(PDF)

## Acknowledgments

The authors would like to thank the participants and collaborators of both studies (the PREFREC and ENSPA studies), as well as the help of Cecilio Niño with the use of statistical software (SPSS) and Emmanuel Ureña with the support in socioeconomic analysis. IMV and HQ belong to the SNI (Spanish language for "National Research System, SENACYT").

## Author Contributions

**Conceptualization:** Angela Isabel Del Rio, Jorge Motta, Hedley K. Quintana.

**Data curation:** Angela Isabel Del Rio.

**Formal analysis:** Angela Isabel Del Rio, Hedley K. Quintana.

**Methodology:** Ilais Moreno Velásquez, Reina Roa, Roger Montenegro Mendoza, Jorge Motta, Hedley K. Quintana.

**Visualization:** Angela Isabel Del Rio.

**Writing – original draft:** Angela Isabel Del Rio.

**Writing – review & editing:** Ilais Moreno Velásquez, Roger Montenegro Mendoza, Jorge Motta, Hedley K. Quintana.

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
