## [Decision Letter · Decision Letter 0]

23 Jun 2022

PONE-D-21-34832Prevalence and diagnosis of hypertension and possible risk factors of undiagnosed hypertension among individuals aged 30–75 years from two Panamanian provinces: Results from population-based cross-sectional studies, 2010 and 2019.PLOS ONE

Dear Dr. Del Rio,

Thank you for submitting your manuscript to PLOS ONE. After careful consideration, we feel that it has merit but does not fully meet PLOS ONE’s publication criteria as it currently stands. Therefore, we invite you to submit a revised version of the manuscript that addresses the points raised during the review process.

We look forward to receiving your revised manuscript.

Kind regards,

Taeyun Kim

Academic Editor

PLOS ONE

“This study was supported by the Ministry of Health of Panama, which provided resources from the selective excise tax on cigarettes and other tobacco products. The funder had no role in study design, data collection and analysis, decision to publish, or preparation of the manuscript.”

Reviewers' comments:

**Comments to the Author**

1. Is the manuscript technically sound, and do the data support the conclusions?

Reviewer #1: Partly

Reviewer #2: Yes

2. Has the statistical analysis been performed appropriately and rigorously? 

Reviewer #1: Yes

Reviewer #2: I Don't Know

3. Have the authors made all data underlying the findings in their manuscript fully available?

Reviewer #1: Yes

Reviewer #2: Yes

4. Is the manuscript presented in an intelligible fashion and written in standard English?

Reviewer #1: Yes

Reviewer #2: Yes

5. Review Comments to the Author

Reviewer #1: The paper actually does not define prevalence of hypertension and undiagnosed hypertension, and Figure 1 makes the situation more confusing. Those with hypertension are defined as having BP ≥ 140/90 mmHg or self-reported hypertension, where the denominator of the prevalence is presumed to be the entire population. For undiagnosed hypertension, the numerator is those without a self-report of a hypertension but with BP ≥ 140/90 mmHg, but the denominator of prevalence estimate is not clear (i.e., is it the entire population or just the population with a self-report of hypertension). The figure shows that approximately half of the population is excluded from the estimation of uncontrolled hypertension, largely because they are “within self-reported medical history and measured SBP/DBP < 140/90.” This suggests that the prevalence of undiagnosed hypertension is restricted to those with a self-report; however, it makes no sense at all to exclude those with controlled hypertension! That the prevalence of hypertension and undiagnosed hypertension in men is 46.5% and 52.3% suggests again that the prevalence of undiagnosed hypertension is restricted to those with a self-reported hypertension (how else could a larger proportion of participants have undiagnosed hypertension than have hypertension). That these prevalence estimates are shifting denominators is not clearly explained and will confuse many readers (perhaps even including this reviewer?).

In a closely related issue, the numbers in the numerator and denominator of the prevalence estimates are not provided in either the text or tables, and would help the reader understand what is being calculated. Please provide the “n”.

The finding of a decreased prevalence of hypertension but an increased prevalence of undiagnosed hypertension is surprising. The populations in the two waves of the assessments are quite different, for example those in the ENSPA had a quite different distribution of education (with a larger number with secondary education, but fewer in either the low or high educated groups), had higher income, were more likely non-urban, and were more likely obese and physically inactive. The paper would be substantially strengthened to see if the differences in the prevalence of hypertension and undiagnosed hypertension are attributable to these differences (which can be assessed in a multivariable model including all data and estimating the odds of positive prevalence after adjustment for the other factors).

Reviewer #2: thank you for your great data about national prevalance of hypertension.

I don't understand that half of the individuals who participated in health check-ups did not have a BP assessment.

It would be helpful to understand better if you add a description of Panama's health screening system.

6. PLOS authors have the option to publish the peer review history of their article (what does this mean?). If published, this will include your full peer review and any attached files.

Reviewer #1: No

Reviewer #2: No

---

## [Author Response · Author response to Decision Letter 0]

23 Jul 2022

Reply to Reviewer #1:

Thank you for taking the time for reading the submitted manuscript. We apologize that in the previous version of the manuscript, the prevalence calculations for hypertension and for undiagnosed hypertension were not defined. To clarify this issue, we have first, changed the term "undiagnosed hypertension" to "unawareness of hypertension" throughout the revised manuscript. 

In our manuscript, hypertension was defined as (a) having a self-reported medical history of hypertension or (b) having a mean SBP ≥140 and/or DBP ≥90 mmHg (regardless of their self-reported medical history of hypertension) or (c) a combination of both. This sample constituted our study population for estimating the prevalence of hypertension unawareness. Unawareness of hypertension was defined as having a mean SBP ≥140 and/or DBP ≥90 mmHg and no self-reported medical history of hypertension.

Following the Reviewer suggestions, we have now:

1. Renamed the title to: 

"Prevalence of hypertension and possible risk factors for hypertension unawareness among individuals aged 30–75 years from two Panamanian provinces: Results from population-based cross-sectional studies, 2010 and 2019."

2. Rewrote the Method section between the lines 153 to 159 as follows: 

"Hypertension was defined as (a) having a self-reported medical history of hypertension or (b) having a mean SBP ≥140 and/or DBP ≥90 mmHg (regardless of their self-reported medical history of hypertension) or (c) a combination of both [27]. This sample constituted our study population for estimating the prevalence of hypertension unawareness. Unawareness of hypertension was defined as having a mean SBP ≥140 and/or DBP ≥90 mmHg and no self-reported medical history of hypertension. We excluded participants who denied self-reported medical history of hypertension and were also missing the second and/or third BP measurement(s) (n=120 in the PREFREC study; n=487 in the ENSPA study) (Fig 1)."

3. Edited the abstract under the Method section as follows:

"Individuals aged 30–75 years who had (a) self-reported history of hypertension or (b) blood pressure (BP) ≥140/90mmHg or (c) a combination or both were classified as hypertensive. Participants with BP≥140/90mmHg who denied a history of hypertension were considered unaware of the condition." 

4. Modified Figure 1; where we wrote down the weighted and unweighted number of participants with hypertension with and without self-reported medical history of hypertension.

We agree with the Reviewer that the point of merging study data further strengthens the manuscript. The authors initially proposed to do so. However, due to different methodologies used in PREFREC 2010 (measuring the blood pressure in the right arm) and ENSPA 2019 (measuring it in the left arm), as we mentioned in Manuscript lines 143–151, we decided not to merge both databases for analysis. According to the literature, BP readings in the right side are higher than left ones.

Instead, our approach was to perform analysis looking for possible risk factors associated with hypertension unawareness in the PREFREC (2010) and ENSPA (2019) study, separately (Table 2); and our point estimates were consistent for many of the factors analyzed. For example, we found similar points estimates for sex (men were more likely to have unawareness) and for blood pressure assessment in the year before study enrollment (protective factor in both studies); whereas age <50 years had a higher point estimate in the ENSPA (2019) study, independent of the same median age in the study populations. 

Furthermore, we found that individuals with no/primary education were more likely to be unaware in the PREFREC (2010) study, but no association was found in the ENSPA (2019) study; this finding could be due to fewer individual reporting no/primary education in the ENSPA (2019) study. Interestingly, in the PREFREC (2010) study, we found decreased odds for unawareness in obese and physical inactive individuals, whereas obesity was not associated to our outcome of interest in ENSPA (2019) study. Other possible factors were discussed in more detail in the Discussion section.

Despite the decision not to show the merge data from both studies, for editorial purposes, we performed logistic regression models to assess the change over time of hypertension and its unawareness with the adjustments for the model A and B shown in Manuscript Table 2, using the PREFREC study as the reference. Hypertension decreased 20% between both studies with borderline significance (Model A= OR: 0.82 [95% CI: 0.64–1.04]; Model B= 0.79 [0.62–1.02]). On the other hand, hypertension unawareness increased between 70-80% during the same period (Model A= OR: 1.81 [95% CI: 1.25–2.63]; Model B= 1.70 [1.15–2.51]). 

Reply to Reviewer #2:

The authors thank you for your comment. Panama's health screening system is based on national technical-administrative standards published in 2018. According to these standards, blood pressure should be screened on any patient attending a healthcare facility aged 18 years or older. However, we found a lack of adherence to the aforementioned standards, since half of the participants stated that attended an annual check-up denied having blood pressure measures taken in the same period of time. 

Following your suggestion, we added the following statement in the revised manuscript (Lines 348 and 349) to make clear the Panamanian standards on hypertension screening:

"Of note, the current national healthcare program states that BP levels should be measured in the right arm at every health check-up in patients 18 years or older."

THE COMPLETE RESPONSE TO REVIEWERS IS ATTACH IN THE "ATTACH FILES" SECTION.

---

## [Decision Letter · Decision Letter 1]

11 Aug 2022

PONE-D-21-34832R1Prevalence of hypertension and possible risk factors of hypertension unawareness among individuals aged 30–75 years from two Panamanian provinces: Results from population-based cross-sectional studies, 2010 and 2019.PLOS ONE

Dear Dr. Del Rio,

Thank you for submitting your manuscript to PLOS ONE. After careful consideration, we feel that it has merit but does not fully meet PLOS ONE’s publication criteria as it currently stands. Therefore, we invite you to submit a revised version of the manuscript that addresses the points raised during the review process.

We look forward to receiving your revised manuscript.

Kind regards,

Taeyun Kim

Academic Editor

PLOS ONE

Reviewers' comments:

Reviewer's Responses to Questions

**Comments to the Author**

1. If the authors have adequately addressed your comments raised in a previous round of review and you feel that this manuscript is now acceptable for publication, you may indicate that here to bypass the “Comments to the Author” section, enter your conflict of interest statement in the “Confidential to Editor” section, and submit your "Accept" recommendation.

Reviewer #1: (No Response)

Reviewer #2: All comments have been addressed

2. Is the manuscript technically sound, and do the data support the conclusions?

Reviewer #1: Yes

Reviewer #2: Yes

3. Has the statistical analysis been performed appropriately and rigorously? 

Reviewer #1: Yes

Reviewer #2: Yes

4. Have the authors made all data underlying the findings in their manuscript fully available?

Reviewer #1: Yes

Reviewer #2: Yes

5. Is the manuscript presented in an intelligible fashion and written in standard English?

Reviewer #1: Yes

Reviewer #2: Yes

6. Review Comments to the Author

Reviewer #1: The attempt of the authors to be responsive to the previous suggestions are appreciated, particularly the specificity in providing the definitions for hypertension and unawareness of hypertension. The authors also followed the suggestion to provide multivariable modeling for predictors of hypertension; however, unfortunately this reviewer must not have been clear in the suggestion. The lead sentence of the discussion is “… findings indicate a decline in the prevalence of hypertension between the PREFREC (2010) and 299 ENSPA (2019) studies, and an increase in the prevalence of hypertension unawareness.” The discussion follows that this may be attributable changes in exposures including increases in obesity and changes in urban/rural status. The difference could also be potentially attributed to including a lower education in ENSPA than PREFREC and much higher levels of physical inactivity. The suggestion that was apparently not clear in the previous review was to consider one multivariable model that included the data from both surveys, where the prevalence of hypertension and the prevalence of unawareness of hypertension is predicted with the period/survey as a predictor variable. This would allow the assessment of whether the odds of hypertension, and the odds of unawareness of hypertension, was different between the two surveys after adjustment for the changes in the population characteristics. While the paper is interesting as it stands, this would directly assess if there are differences in the prevalence of hypertension, and in the prevalence of unawareness of hypertension, after adjustment for differences in the individuals in the two studies.

Reviewer #2: I am very happy to be able to do this review. It's a very interesting topic and I think it's a logical conclusion overall.

7. PLOS authors have the option to publish the peer review history of their article (what does this mean?). If published, this will include your full peer review and any attached files.

Reviewer #1: No

Reviewer #2: No

---

## [Author Response · Author response to Decision Letter 1]

16 Aug 2022

Reply to Reviewer #1: 

Thank you for the time invested in revising and improving the manuscript. 

Following the Reviewer's suggestions, the authors now support the claim that there were changes of hypertension prevalence and its awareness between both studies. We added multivariable logistic regression models of the prevalence of hypertension and hypertension unawareness using merged data from both studies:

In consequence, the following fragment under the statistical analysis heading was added: 

"In addition, we included in the Supporting Information (S2 and S3 Tables) crude and multivariable logistic regression models using merged data from both studies (both PREFREC and ENSPA) in which the prevalence of hypertension (S2 Table) and the prevalence of hypertension unawareness (S3 Table) were the outcome variables, respectively, with the aforementioned predictor variables together with the study period as an additional predictor (being PREFREC the reference level)."

The following fragment was added to the results section:

"As shown in the S2 and S3 Tables, after merging all the data from both studies and adjusting for all other variables, there was a 21% borderline decrease in the prevalence of hypertension (ENSPA OR: 0.79 [95% CI: 0.62–1.02], PREFREC as reference), but a 70% increase in hypertension unawareness (ENSPA OR: 1.70 [95% CI: 1.15–2.51], PREFREC as reference), respectively." 

Reply to Reviewer #2: 

The authors are grateful for your time and interest in publishing the manuscript.

---

## [Decision Letter · Decision Letter 2]

22 Aug 2022

PONE-D-21-34832R2Prevalence of hypertension and possible risk factors of hypertension unawareness among individuals aged 30–75 years from two Panamanian provinces: Results from population-based cross-sectional studies, 2010 and 2019.PLOS ONE

Dear Dr. Del Rio,

Thank you for submitting your manuscript to PLOS ONE. After careful consideration, we feel that it has merit but does not fully meet PLOS ONE’s publication criteria as it currently stands. Therefore, we invite you to submit a revised version of the manuscript that addresses the points raised during the review process.

We look forward to receiving your revised manuscript.

Kind regards,

Taeyun Kim

Academic Editor

PLOS ONE

Journal Requirements:

Additional Editor Comments (if provided):

Dear authors.

I appreciate your great efforts in this work.

I believe this study is almost ready for publication.

Please amend your comment regarding the statistical statement about the OR, which was raised from a Reviewer.

Reviewers' comments:

Reviewer's Responses to Questions

**Comments to the Author**

1. If the authors have adequately addressed your comments raised in a previous round of review and you feel that this manuscript is now acceptable for publication, you may indicate that here to bypass the “Comments to the Author” section, enter your conflict of interest statement in the “Confidential to Editor” section, and submit your "Accept" recommendation.

Reviewer #1: (No Response)

2. Is the manuscript technically sound, and do the data support the conclusions?

Reviewer #1: Yes

3. Has the statistical analysis been performed appropriately and rigorously? 

Reviewer #1: Yes

4. Have the authors made all data underlying the findings in their manuscript fully available?

Reviewer #1: Yes

5. Is the manuscript presented in an intelligible fashion and written in standard English?

Reviewer #1: Yes

6. Review Comments to the Author

Reviewer #1: Thank you for adding the requested analysis, it does allow for the quantification of whether temporal changes are present. My only additional comment is that you note " ... there was a 21% borderline decrease in the prevalence of hypertension (ENSPA OR: 0.79 [95% CI: 0.62–1.02], PREFREC as reference), but a 70% increase in hypertension unawareness (ENSPA OR: 1.70 [95% CI: 1.15–2.51], PREFREC as reference), respectively." This is an incorrect statement, as logistic regression provides odds ratios (not relative risks), so it needs to state that the odds (not prevelence) of hypertension were 21% lower and 70% higher respectively.

7. PLOS authors have the option to publish the peer review history of their article (what does this mean?). If published, this will include your full peer review and any attached files.

Reviewer #1: No

---

## [Author Response · Author response to Decision Letter 2]

8 Sep 2022

We again thank you for the time invested in revising and improving the manuscript. 

Following the Reviewer's suggestion, the Results and the Discussion sections were rewritten to reflect that logistic regression models represent changes in the adjusted odds between ENSPA and PREFREC studies. 

Lines 251 to 255 from the Results section were edited as follows: 

"As shown in the S2 and S3 Tables, after merging all data from both studies and adjusting for all other variables, there was a 21% borderline decrease in the odds of having hypertension in the ENSPA (2019) study compared to the PREFREC study (ENSPA OR: 0.79; 95% CI: 0.62–1.02), but a 70% increase in the odds of being unaware (ENSPA OR: 1.70; 95% CI: 1.15–2.51), respectively."

Lines 306 to 307 from the Discussion section were edited as follows: 

"Our findings indicate a borderline decrease in the odds of having hypertension between the PREFREC (2010) and ENSPA (2019) studies, but an increase in the odds of being unaware of hypertension."

Journal Requirements:

The Reference list was revised at the request of the journal. All references were complete and correct, except reference 52 which has been updated, but did not alter the analysis of our discussion. Therefore, no changes have been made to the reference list.

---

## [Editor Report · Decision Letter 3]

4 Oct 2022

Prevalence of hypertension and possible risk factors of hypertension unawareness among individuals aged 30–75 years from two Panamanian provinces: Results from population-based cross-sectional studies, 2010 and 2019.

PONE-D-21-34832R3

Dear Dr. Del Rio,

We’re pleased to inform you that your manuscript has been judged scientifically suitable for publication and will be formally accepted for publication once it meets all outstanding technical requirements.

Kind regards,

Taeyun Kim

Academic Editor

PLOS ONE
---

## [Editor Report · Acceptance letter]

11 Oct 2022

PONE-D-21-34832R3 

Prevalence of hypertension and possible risk factors of hypertension unawareness among individuals aged 30–75 years from two Panamanian provinces: Results from population-based cross-sectional studies, 2010 and 2019. 

Dear Dr. Del Rio:

I'm pleased to inform you that your manuscript has been deemed suitable for publication in PLOS ONE. Congratulations! Your manuscript is now with our production department. 

Kind regards, 

on behalf of

Dr. Taeyun Kim 

Academic Editor

PLOS ONE